# GraphVis: Boosting LLMs with Visual Knowledge Graph Integration

**Yihe Deng**[*]    **Chenchen Ye**    **Zijie Huang**
**Mingyu Derek Ma**    **Yiwen Kou**    **Wei Wang**

University of California, Los Angeles

## Abstract

The rapid evolution of large language models (LLMs) has expanded their capabilities across various data modalities, extending from well-established image data to increasingly popular graph data. Given the limitation of LLMs in hallucinations and inaccuracies in recalling factual knowledge, Knowledge Graph (KG) has emerged as a crucial data modality to support more accurate reasoning by LLMs. However, integrating structured knowledge from KGs into LLMs remains challenging, as most KG-enhanced LLM methods directly convert the KG into linearized text triples, which is not as expressive as the original structured data. To address this, we introduce `GraphVis`, which conserves the intricate graph structure through the visual modality to enhance the comprehension of KGs with the aid of Large Vision Language Models (LVLMs). Our approach incorporates a unique curriculum fine-tuning scheme which first instructs LVLMs to recognize basic graphical features from the images, and subsequently incorporates reasoning on QA tasks with the visual graphs. This cross-modal methodology not only markedly enhances performance on standard textual QA but also shows improved zero-shot VQA performance by utilizing synthetic graph images to augment the data for VQA tasks. We present comprehensive evaluations across commonsense reasoning QA benchmarks, where `GraphVis` provides an average improvement of $11.1\%$ over its base model and outperforms existing KG-enhanced LLM approaches. Across VQA benchmarks such as ScienceQA that share similar scientific diagram images, `GraphVis` provides a notable gain of $4.32\%$. Code is made available on GitHub.

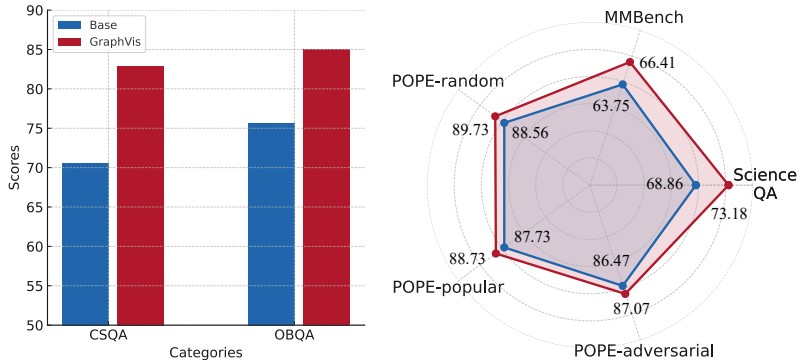

Figure 1: **Left**: Accuracy improvement of `GraphVis` compared to the base model's performance on commonsense reasoning tasks. **Right**: Improvement by `GraphVis` on multiple VQA benchmarks over its base LVLM model LLaVA-v1.6 (Liu et al., 2024).

---

[*]Corresponding to Yihe Deng <yihedeng@cs.ucla.edu>.

38th Conference on Neural Information Processing Systems (NeurIPS 2024).

# 1 Introduction

The rapid evolution of large language models (LLMs) (Zhang et al., 2019; Brown et al., 2020; Touvron et al., 2023a; Chung et al., 2024) has unlocked new opportunities for interacting with multimodal data sources. Approaches that enable input from multi-modal data can expands the information that LLMs can take in for various downstream reasoning tasks across domains. The modalities in existing LLM-based models span across images (Zhu et al., 2023; Liu et al., 2023b), videos (Maaz et al., 2023; Li et al., 2023c), and audio (Zhang et al., 2023; Rubenstein et al., 2023). Most recently, researchers have also begun to build an unified architecture to encode diverse modalities jointly (Wu et al., 2023). Such unification holds considerable promise, and poses an interesting question on whether data from one modality could enhance the model performance in another.

Beyond the frequently explored modalities such as vision and audio, knowledge graphs (KGs) are also gaining attention. Given LLMs' limitations such as hallucinations (Li et al., 2023a), inaccuracies in recalling factual knowledge (Yang et al., 2023), and the costly updates of knowledge via training (Ding et al., 2023), researchers are exploring KGs as a robust source of structured and easy-to-update facts (Pan et al., 2023; Jin et al., 2023; Agrawal et al., 2023; Huang et al., 2023). The use of KGs to enhance language models began with smaller models like BERT (Kenton and Toutanova, 2019; Huang et al., 2022), incorporating KGs into the pre-training objectives (Zhang et al., 2019; Rosset et al., 2020; Wang et al., 2021) or integrating them through architectural modifications (Yasunaga et al., 2021; Zhang et al., 2022). However, the recent development of larger and more complex LLMs poses challenges in adapting these earlier methods. Current strategies for integrating KGs into LLMs fall into two categories: (1) verbalizing relevant KG triples and directly appending them to prompts as "(node a, edge, node b)" (Guo et al., 2023; Feng et al., 2023; Baek et al., 2023) or (2) employing a graph neural network (GNN) to generate embeddings for relevant KG subgraphs and projecting these into the LLM's token embedding space (Chai et al., 2023; Tian et al., 2024). Nonetheless, these approaches often yield results that are either weaker than or comparable to methods that fully fine-tune smaller LMs with integrated KG information, revealing an underutilization of the graph structure and rich relational context. Thus, effectively integrating the KG modality into LLMs and enabling them to comprehend graph concepts remains an unsolved challenge.

With the rapid advancement of LLMs across various modalities, a question arises: can multimodal LLMs, trained in domains other than KGs, facilitate the understanding of graph structures? Large Vision Language Models (LVLMs) (Liu et al., 2023b), pre-trained on an extensive corpus of image-text pairs, demonstrate exceptional abilities in processing image inputs. In response to this potential, we introduce a novel methodology, `GraphVis`, that enhances graph comprehension by visualizing subgraphs and leveraging LVLMs for KG-enhanced question answering. This approach involves translating retrieved subgraphs into images, which are then processed by an LVLM to aid in answering questions. Recognizing that LVLMs typically lack proficiency with visual graphs, we design a unique curriculum fine-tuning scheme. Initially, the model is trained to interpret simple graphical features, such as node count, edge count, and node degree, through self-supervised learning. It then progresses to handling more complex queries by integrating textual question-answer data with relevant visualized subgraphs, thereby fine-tuning the LVLM to respond accurately to KG-based questions using these images. Our experiments demonstrate that this approach effectively improves the model's performance on downstream QA tasks, surpassing both current KG-enhanced LLMs and traditional fully fine-tuned KG language models.

While the vision modality significantly enhances the integration of KGs and improves the performance on textual QA tasks, our study extends this exploration to the benefits of KGs and textual QA data for LVLMs in zero-shot visual question-answering (VQA) tasks. Notably, images resembling graphs are prevalent in current VQA tasks (Lu et al., 2022; Yu et al., 2023a; Lu et al., 2024), yet similar training datasets are scarce. Our research demonstrates that the availability of extensive textual QA data and relevant KGs facilitates the generation of large synthetic datasets that feature graph images, effectively addressing this scarcity and supporting the training of LVLMs on such data. Evaluations across multiple VQA benchmarks reveal that our LVLM, fine-tuned with the `GraphVis` approach, also shows remarkable improvements in VQA performance.

Our contributions are summarized as follows:

- We introduce a novel method, `GraphVis`, that employs visual modality to enhance the understanding of KGs in LLMs, leveraging graph visualization to bridge the gap between structured KG data and multimodal LLM processing capabilities.

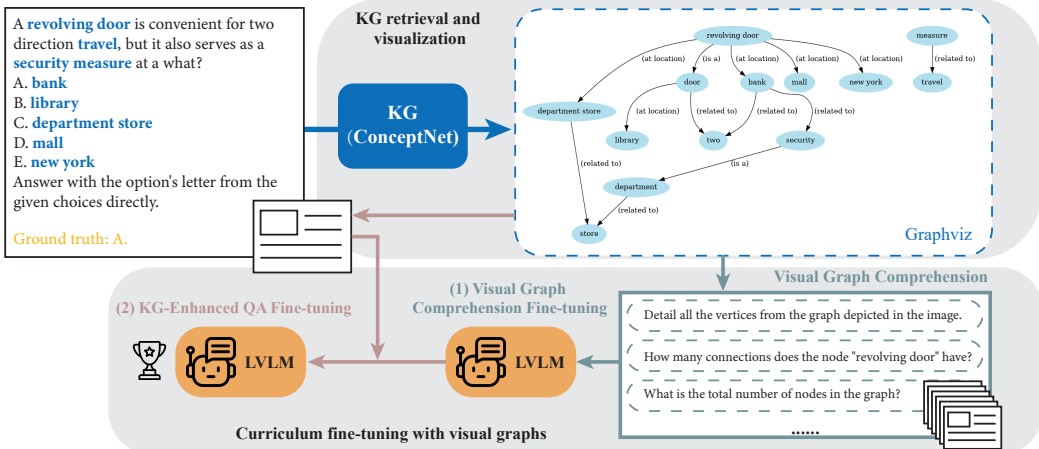

Figure 2: Overview of `GraphVis`. Given an input question and answer pair in the training data, we retrieve and visualize the relevant subgraph. With pre-defined questions on the basic features of the visual graphs such as numbers of nodes and node degree, we first construct data for visual graph comprehension fine-tuning. Subsequently, we incorporate the QA pair with the visual graph for KG-enhanced QA fine-tuning.

- We present a unique curriculum fine-tuning scheme tailored for LVLMs that sequentially trains on graph-derived images first to visual graph comprehension and then to apply this understanding in more complex QA contexts.
- We offer a new perspective on gathering fine-tuning data to enhance LVLMs. Specifically, we propose that pure textual data can be combined with relevant synthetic graph images derived from KGs to improve the LVLM's capability in image comprehension and reasoning.

## 2 Related Work

**KG-Enhanced LLMs.** Initial studies on KG-enhanced language models have shown that integrating KGs into the pre-training objectives can enrich the foundational knowledge of language models. This approach has been largely applied to encoder-only language models such as BERT (Kenton and Toutanova, 2019) with training objectives specifically tailored for these models such as masked word prediction (Zhang et al., 2019; Shen et al., 2020; Zhang et al., 2020; Rosset et al., 2020; Wang et al., 2021; Li et al., 2022; Kang et al., 2022; Baek et al., 2023). Another line of work also relies on the encoder architecture of language models and performs full-parameter fine-tuning on a KG encoder for the fusion of knowledge (Sun et al., 2021; Yasunaga et al., 2021; Zhang et al., 2022; Yujie et al., 2023). However, the advent of recent decoder-only LLM pre-trained on a significantly larger scale (e.g., the GPT series (Radford et al., 2019; Brown et al., 2020; OpenAI, 2023), LLaMA series (Touvron et al., 2023a,b) and Mistral (Jiang et al., 2023)), increases the difficulty and cost of adapting these KG-based pre-training and fine-tuning methods to current LLMs. Consequently, with KGs as a distinct modality, researchers have been exploring various methods for integrating the information into LLMs. One straightforward and most commonly used approach involves verbalizing relevant knowledge graphs and appending them to the prompts (Guo et al., 2023; Feng et al., 2023; Fatemi et al., 2023; Sun et al., 2023; Luo et al., 2023). For a notable example, KAPING (Baek et al., 2023) retrieves the top $k$ most relevant knowledge triples to the prompt and appends it in the form of textual triples to the original prompt. Meanwhile, such an approach linearizes the originally structured information and does not maintain a natural language form. Another research direction therefore employs GNNs to generate embeddings for retrieved subgraphs, subsequently projecting these into the LLM's token embedding space as soft prompts to preserve structured graph information (Yasunaga et al., 2021; Hu et al., 2022; Zhang et al., 2022; Chai et al., 2023; Tian et al., 2024). Nevertheless, GNN-based approaches require task-specific fine-tuning and may struggle with generalization across new tasks.

**Multimodal LLMs.** Other than incorporating knowledge graphs, popular investigations on multimodal inputs to LLMs include image (Zhu et al., 2023; Liu et al., 2023b), video (Maaz et al., 2023; Li et al., 2023c), audio (Zhang et al., 2023; Rubenstein et al., 2023) and temporal data (Yu et al., 2023b; Chang et al., 2023). Significantly, advances in pre-trained vision-language models (Radford et al., 2021; Jia et al., 2021; Alayrac et al., 2022), which align the visual and textual embedding spaces on web-scale image-caption data, have facilitated substantial progress in the development of Large

Vision Language Models (LVLMs) (Liu et al., 2023a,b; Zhu et al., 2023; Chen et al., 2023; Ye et al., 2023; Dai et al., 2023; Gao et al., 2023; Bai et al., 2023; Peng et al., 2023). These models, with vision encoders trained on extensive collections of web images, exhibit robust visual reasoning capabilities across a range of tasks (Gao et al., 2015; Lu et al., 2022; Xu et al., 2023; Lu et al., 2024). However, the incorporation of image data with graph structures into both pre-training and benchmark datasets remains limited, primarily appearing as scientific diagrams within visual question answering datasets for mathematics and science (Lu et al., 2022, 2024). This paper also sheds light on an interesting potential to acquire a large volume of graph images through text-based QA datasets to enhance the capabilities of LVLMs.

## 3 Problem Setting and Preliminaries

**Notation.** We use lower case letters to denote scalars and lower case bold face letters to denote vectors. We denote an input sequence, or prompt, as $\mathbf{x} = [x_1, \ldots, x_n]$, where $x_i$ represents a token in the LLM's vocabulary. Then, we use the symbol $p(\cdot|\mathbf{x})$ to represent the conditional probability of LLM's response given the prompt $\mathbf{x}$. Lastly, we denote the sequence of tokens generated before the $t$-th token as $\mathbf{y}_{<t} = [y_1, \ldots, y_{t-1}]$ for $t > 1$.

**Generative Language Models.** Let $p_{\boldsymbol{\theta}}$ denotes an LLM parameterized by $\boldsymbol{\theta}$. We consider a sequence $\mathbf{x} = [x_1, \ldots, x_n]$ as the input prompt, for which each $x_i$ is a token from the LLM's vocabulary. The LLM then generates the response sequence $\mathbf{y} = [y_1, \ldots, y_m]$ by sampling from the conditional probability distribution $p_{\boldsymbol{\theta}}(\cdot|\mathbf{x})$, where $y_t$ denotes individual token for $1 \leq t \leq m$. The conditional distribution $p_{\boldsymbol{\theta}}(\mathbf{y}|\mathbf{x})$ can therefore be expressed as a Markov process $p_{\boldsymbol{\theta}}(\mathbf{y}|\mathbf{x}) = \prod_{t=1}^{m} p_{\boldsymbol{\theta}}(y_t|\mathbf{x}, \mathbf{y}_{<t})$. Given a supervised fine-tuning dataset, $S = \{(\mathbf{x}, \mathbf{y})\}_{i=1}^{n}$, the training objective is therefore to maximize the model's likelihood of generating $\mathbf{y}$ given $\mathbf{x}$, resulting in the following loss function:

$$L(\boldsymbol{\theta}) = \mathbb{E}_{(\mathbf{x}, \mathbf{y}) \sim S} \Big[ -\log p_{\boldsymbol{\theta}}(\mathbf{y}|\mathbf{x}) \Big]. \tag{3.1}$$

Given an LLM, an LVLM additionally contains two more components, including a vision encoder $f_v(\cdot)$ and a projection network $f_p(\cdot)$. The model processes an additional image input $\mathbf{e}$, which is converted into visual tokens within the language token space by the vision encoder and the projection network, producing $\mathbf{v} = [v_1, \ldots, v_k] = f_v \circ f_p(\mathbf{e})$. The conditional probability distribution $p_{\boldsymbol{\theta}}(\mathbf{y}|\mathbf{v}, \mathbf{x})$ is thus decomposed as

$$p_{\boldsymbol{\theta}}(\mathbf{y}|\mathbf{v}, \mathbf{x}) = \prod_{j=1}^{m} p_{\boldsymbol{\theta}}(y_j|\mathbf{v}, \mathbf{x}, \mathbf{y}_{<j}). \tag{3.2}$$

**KG-enhanced LLMs.** A knowledge graph, denoted as $\mathcal{G} = \{\mathcal{V}, \mathcal{E}\}$, consists of a set of vertices $\mathcal{V}$ and their connections, or edges, $\mathcal{E}$. Considering an input question $\mathbf{x} = [x_1, \ldots, x_n]$ with its corresponding ground truth answer $\mathbf{y}^*$, we define $\mathcal{V}_{\mathbf{x}} = \{\mathbf{v}_i\}_{i \in \mathcal{I}_{\mathbf{x}}} \subseteq \mathcal{V}$ as the vertices mentioned in $\mathbf{x}$, where $\mathcal{I}_{\mathbf{x}}$ is the index set of vertices associated with the tokens in the question. The objective of KG-enhanced LLM can be decomposed into two steps: (1) relevant subgraph retrieval and (2) effective knowledge projection to the language embedding space. Subgraph retrieval involves designing a function $f$ that generates a subgraph most relevant to the input prompt and containing the mentioned vertices $\mathcal{V}_{\mathbf{x}}$ and connected via the edges $\mathcal{E}_{\mathbf{x}}$:

$$f(\mathbf{x}, \mathcal{V}_{\mathbf{x}}, \mathcal{G}) = \{\mathcal{V}_{\mathbf{x}}, \mathcal{E}_{\mathbf{x}}\} = \mathcal{G}_{\mathbf{x}} \subset \mathcal{G}.$$

The function $f$ could be pre-defined or trained. In this work, we consider the same approach as previous works (Feng et al., 2020), where $\mathcal{E}_{\mathbf{x}}$ is obtained from all k-hop paths connecting two nodes in $\mathcal{V}_{\mathbf{x}}$. Given a relevant subgraph $\mathcal{G}_{\mathbf{x}}$, the target of effectively leveraging the information is to construct a function $g$ that generates informative tokens such that

$$p_{\boldsymbol{\theta}}(\mathbf{y}^*|\mathbf{x}^g) = \max_{\mathbf{x}} p_{\boldsymbol{\theta}}(\mathbf{y}^*|\mathbf{x}),$$

where $\mathbf{x}^g = [g(\mathcal{G}_{\mathbf{x}}), \mathbf{x}]$ is the KG-augmented prompt. In essence, the function $g$ finds a way of leveraging the knowledge graph to enhance the language model's capacity for answering questions. The current methods therefore fall into the framework as

- Linearize. The linearization process is to represent the KG as a list of triples: $g(\mathcal{G}_{\mathbf{x}}) = [(\mathbf{v}_1, \mathbf{e}_1, \mathbf{u}_1), (\mathbf{v}_2, \mathbf{e}_2, \mathbf{u}_2), \cdots]$ where edge $\mathbf{e}_i \in \mathcal{E}_{\mathbf{x}}$ and $\mathbf{v}_i, \mathbf{u}_i$ are two endpoints of $\mathbf{e}_i$.
- GNN-based. The GNN-based methods leverage a GNN model for the additional information: $\mathbf{x}^g = [g_{\text{GNN}}.(\mathcal{G}_{\mathbf{x}}), \mathbf{x}]$.

# 4 Method

In this section, we formally introduce `GraphVis`, a technique that employs LVLMs to enhance the integration of KG information, thereby improving performance in downstream textual QA tasks. Reversely, `GraphVis` also enhances the performance of LVLMs in visual QA tasks by utilizing extensive data from both textual and KG modalities. The methodology of `GraphVis` is outlined in Algorithm 1 and demonstrated in Figure 2. We further elaborate the details of the method below.

`GraphVis` consists of two major components: (1) a novel integration of the retrieved subgraph for KG-enhanced QA via visualization of the graph, and (2) a progressive fine-tuning approach that starts by understanding graphical features and subsequently leverages them for reasoning. The primary objective of `GraphVis` is to improve the incorporation of KG information into LVLMs rather than enhancing retrieval techniques. Therefore, we adopt the same subgraph retrieval approach as previous studies (Lin et al., 2019; Feng et al., 2020; Yasunaga et al., 2021), which involves retrieving $k$-hop paths between entities mentioned in the input prompts from the entire KG. For visualization, we utilize the Graphviz tool (Gansner and North, 2000) to generate visual representations for each retrieved KG subgraph.

Most importantly, `GraphVis` employs a unique curriculum fine-tuning approach specifically designed for visual graph comprehension. While current LVLMs are fine-tuned on human-labeled vision-language instruction data, images of complex graph structures are much more scarce compared to the many natural images. The reasoning tasks designed for complex graph images are also very limited. GraphVis highlights the potential to leverage textual data and KG images to improve the LVLM's capability in reasoning with graph images. To address this, we initiate the fine-tuning process with simple, self-constructed questions about the structural and relational information in the graph, paired with automatically derived answers, training the model to thoroughly understand visual graphs before progressing to more complex reasoning tasks. The loss objective remains the same as SFT objective (3.1). These questions include,

- **Node description**: name all nodes appeared in the image.
- **Node degree detection**: answer with the degree of a named node in the image.
- **Highest node degree detection**: answer with name(s) of the node(s) that has the highest degree in the image.
- **Node number detection**: answer with the total number of nodes appeared in the image.
- **Edge number detection**: answer with the total number of edges appeared in the image.
- **Triple listing**: describe the image by listing all triples that appeared in the image.

For each of the question types, we draw a prompt from a pool of five pre-defined prompts of the task to add variance to the data. After the model fully understands the features of a visual graph, we proceed to further fine-tune its ability to reason with the visual graph, enhancing its capability to respond to related queries. The original question from the textual QA training dataset is then augmented with the visual subgraph as the following,

```
<visual subgraph>
The image represents a knowledge graph relevant to the question,
which may or may not be useful.  Question:  <original question>
```

The ground truth answers remain unchanged from the textual QA training data. This KG-enhanced QA fine-tuning subsequently starts from the model weights learned in the previous visual graph comprehension fine-tuning phase.

# 5 Experiments

In this section, we present experiment results of `GraphVis` on enhancing commonsense reasoning tasks with retrieved KG subgraphs from ConceptNet, as well as improving the zero-shot VQA capability of the LVLM by leveraging the data from the textual and KG modality. Across several benchmark datasets, we demonstrate the effectiveness of `GraphVis`.

## 5.1 Experiment Setup

**Model and Datasets.** In experiments, we consider `llava-v1.6-mistral-7b` (Liu et al., 2023a) as our base VLM model. We consider ConceptNet (Speer et al., 2017), a commonsense knowledge graph, as the KG used in our experiments. There are 799,273 nodes and 2,487,810 edges in total existing in the KG, and there are 42 specific different types of relations, merged into 17 relations (Feng et al., 2020). In both fine-tuning stage and inference stage, we consider retrieving 2-hop subgraphs for the conciseness of the images while preserving important information. We then

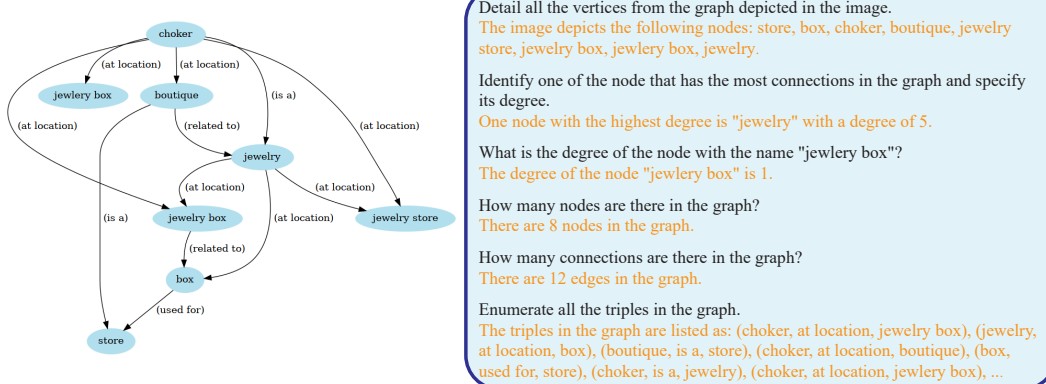

Figure 3: Example of the visual graph comprehension question and answer pairs.

---

**Algorithm 1** `GraphVis`

---

**Input:** Training data from the textual QA dataset: $\{\mathbf{x}^{(i)}, \mathbf{y}^{(i)}\}_{i \in [N]}$. LVLM parameterized by $\boldsymbol{\theta}$: $p_{\boldsymbol{\theta}}$. The relevant KG: $\mathcal{G} = \{\mathcal{V}, \mathcal{E}\}$. Relevent subgraph retrieval method $f$. Self-supervised graph question set $P = \{\mathbf{p}^{(i)}\}_{i \in [M]}$.

Let graphical feature training dataset $D_g = \{\}$ and the graph VQA dataset $D_v = \{\}$.

**for** $i = 1, \dots N$ **do**

    Retrieve the relevant KG subgraph $\mathcal{G}_i = f(\mathbf{x}^{(i)}, \mathcal{V}_{\mathbf{x}^{(i)}}, \mathcal{G})$.

    Plot the KG subgraph to obtain the image for visualized KG $\mathbf{v}^{(i)}$.

    **for** $j = 1, \dots M$ **do**

        Given $\mathbf{p}^{(j)}$ and $\mathcal{G}_i$, automatically get answer $\mathbf{a}^{(j)}$.

        Add $\left(\mathbf{v}^{(i)}, \mathbf{p}^{(j)}, \mathbf{a}^{(j)}\right)$ to $D_g$.

    **end for**

    Add $\left(\mathbf{v}^{(i)}, \mathbf{x}^{(i)}, \mathbf{y}^{(i)}\right)$ to $D_v$.

**end for**

Graph understanding fine-tuning: update $\widehat{\boldsymbol{\theta}} = \operatorname{argmin}_{\boldsymbol{\theta} \in \boldsymbol{\Theta}} \sum_{(\mathbf{v}, \mathbf{x}, \mathbf{y}) \in D_g} \left( -\log p_{\boldsymbol{\theta}}(\mathbf{y}|\mathbf{v}, \mathbf{x}) \right)$.

KG-enhanced QA fine-tuning: update $\widehat{\boldsymbol{\theta}} = \operatorname{argmin}_{\widehat{\boldsymbol{\theta}} \in \boldsymbol{\Theta}} \sum_{(\mathbf{v}, \mathbf{x}, \mathbf{y}) \in D_v} \left( -\log p_{\widehat{\boldsymbol{\theta}}}(\mathbf{y}|\mathbf{v}, \mathbf{x}) \right)$.

**Output:** $\widehat{\boldsymbol{\theta}}$.

---

consider Commonsense QA (CSQA) (Talmor et al., 2019) and OpenBook QA (OBQA) (Mihaylov et al., 2018) as the commonsense reasoning tasks that can be improved via relevant subgraphs in ConceptNet. For the zero-shot VQA tasks, we consider ScienceQA (Lu et al., 2022), MMBench (Liu et al., 2023c) and POPE (Li et al., 2023b) that share similar images or tasks as our synthetic data from textual QA with visual KG subgraphs. Specifically, ScienceQA focuses on scientific question answering and contains scientific diagrams. MMBench is a recent multi-modal benchmark that comprehensively evaluates a model's capabilities in a wide range of tasks and evaluation criteria. POPE evaluates the extent of object hallucinations for LVLMs, formulating a binary classification task by prompting the model with questions such as "Is there an <object> in this image?". For VQA benchmarks, we use the evaluation scripts provided by LLaVA (Liu et al., 2023a) to obtain the results for both our base model and after using `GraphVis` to ensure a fair comparison. In Figure 4 and 5, we demonstrate the statistics of the synthetic visual knowledge graphs in CSQA.

**Baselines.** We consider the previous KG-enhanced methods that fine-tune language models on the training data with ConceptNet as one category of the baselines, including the popular *QA-GNN* (Yasunaga et al., 2021) and *GreaseLM* (Zhang et al., 2022). We further include the performance of current LLMs without KG or fine-tuning, including *FLAN-T5-xxlarge* (11B) (Chung et al., 2024), which is the base LLM used for many KG-enhanced methods, and *GPT-4*. Lastly, we include the reported values of methods on KG-enhanced LLMs including *KAPING* (Baek et al., 2023), *KSL* (Feng et al., 2023) and Graph Neural Promping (*GNP*) (Tian et al., 2024), which all share the same setting of using ConceptNet for enhancement on commonsense reasoning tasks. In particular, KSL and GNP are fine-tuning approaches and we report their best performances (e.g. for GNP, we consider the results from both fine-tuning GNN and projection network and Low-Rank Adaptation (LoRA) (Hu

et al., 2021) fine-tuning on LLM). Lastly, we note that these methods have not open-sourced their codes and models, and therefore we consider our re-implementation of KAPING based on the same VLM as a reference.

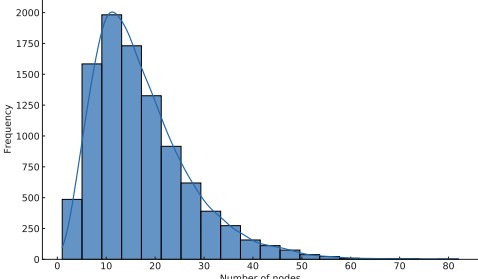

Figure 4: Distribution of node number in CSQA.

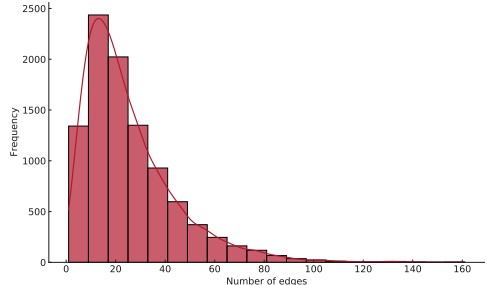

Figure 5: Distribution of edge number in CSQA.

## 5.2 Main Results

In Table 1, we present the main results of `GraphVis` on KG-enhanced question answering. `GraphVis` demonstrates a significant improvement in accuracy over the base model, with an increase of 12.3% on CSQA and 9.9% on OBQA. We include the full fine-tuning methods of KG-enhanced LMs as strong baselines, which include well-designed model architectures based on small-scale language models to better integrate KG information. Although the current methods proposed for KG-enhanced LLMs are not open-sourced at the time of this manuscript, we incorporate the reported values of the baselines, including KSL, KAPING, and GNP. We observe that, due to the strong performance of the base LLMs, prompting methods like KAPING can actually harm performance by causing notably longer contexts with information not in natural language form. Conversely, fine-tuning methods like KSL and GNP offer greater improvements, even though mostly under-performing or matching the performance of full-parameter fine-tuned LMs that have intrinsic architectural changes to adapt the KG information. Meanwhile, the scale of LLMs is unprecedented, causing difficulty in both modifying the architecture or fully fine-tuning all parameters. While `GraphVis` similarly employs LoRA fine-tuning to only update a small amount of parameters similar to KSL and GNP, we observe a much more significant improvement that suggests a better incorporation of the information. On CSQA, `GraphVis` with a 7B LLM surpasses the second-best result, KSL with GPT-3.5 (>100B), by a substantial margin of 3.2%. On OBQA, `GraphVis` remains the top-performing method, outperforming fine-tuning methods like GNP with an 11B LLM by 5.7%.

Table 1: Performance of `GraphVis` compared with the original VLM model across benchmarks and VQA tasks. As current baselines on LLMs are not open-sourced yet, we include the results directly reported from their papers (Zhang et al., 2022; Feng et al., 2023; Tian et al., 2024). We use FT to indicate if a method involves fine-tuning. The **bold** numbers indicate the best results among all methods and underscored numbers represent the second best.

| Category | Method | Base Model | FT | CSQA | OBQA |
|----------|--------|-----------|----|------|------|
| LM | QA-GNN | AristoRoBERTa (355M) | ✓ | 76.1 | 82.8 |
| | GreaseLM | AristoRoBERTa (355M) | ✓ | 78.5 | 84.8 |
| LLM | Base LLM | GPT-3.5 (>100B) | ✗ | 72.9 | 74.8 |
| | KSL | GPT-3.5 (>100B) | ✗ | 79.6 | 81.6 |
| | Base LLM | LLaMA (7B) | ✗ | 38.0 | 29.8 |
| | KSL | LLaMA (7B) | ✓ | 47.4 | 45.8 |
| | Base LLM | FLAN-T5-xxlarge (11B) | ✗ | – | 76.8 |
| | KAPING | FLAN-T5-xxlarge (11B) | ✗ | – | 60.0 |
| | GNP | FLAN-T5-xxlarge (11B) | ✓ | – | 79.8 |
| LVLM | Base LVLM | LLaVA-v1.6-Mistral (7B) | ✗ | 70.5 | 75.6 |
| | KAPING | LLaVA-v1.6-Mistral (7B) | ✗ | 67.7 | 71.2 |
| | GraphVis | LLaVA-v1.6-Mistral (7B) | ✓ | **82.8**$_{(+12.3)}$ | **85.5**$_{(+9.9)}$ |

### 5.2.1 Leveraging KG and Textual Data to Enhance LVLM

Furthermore, we investigate the benefit of `GraphVis` in the reverse direction, by leveraging textual QA dataset and KG to improve the LVLM's zero-shot performance on VQA tasks. We begin with the observation that many prevalent VQA benchmarks, such as ScienceQA Lu et al. (2022), feature images structured as directed graphs. For instance, ScienceQA contains a category of image exists as the food web images with questions to identify the decomposers or the producers in the web, as the example shown in Figure 6a. Similarly, MMBench (Liu et al., 2023c) includes a notable portion of images that comprise charts and diagrams, illustrated in Figure 6b. While current LVLMs are pre-trained and fine-tuned on large corpus of vision-language instruction data, images of graph structures are much more scarce compared to the many natural images, in addition to the scarcity of reasoning tasks designed specifically for graphs. The presence of structured graphical images within VQA benchmarks highlights the potential of `GraphVis` to leverage textual data and KG images to improve the LVLM's capability in reasoning with such type of images.

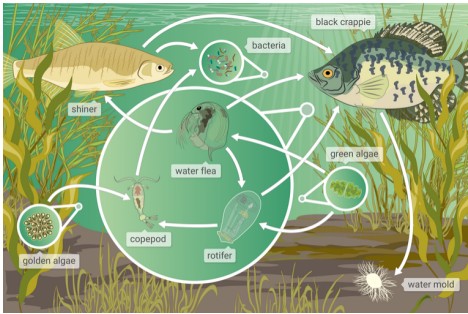

(a) Example image from ScienceQA (Lu et al., 2022). Question: Which of the following organisms is the decomposer in this food web?

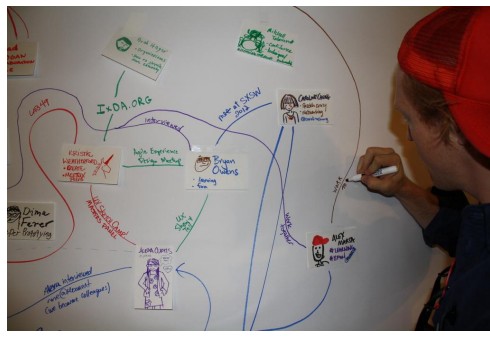

(b) Example image from MM-Bench (Liu et al., 2023c). Question: who is at the center of all of this?

Figure 6: Example images from VQA tasks that share resemblance to the visualized KG subgraphs.

In Table 2, we present the performance of our base LVLM (`llava-v1.6-mistral-7b`) and its comparison after applying `GraphVis`. Notably, `GraphVis` employs synthetic images with textual QAs and does not require human-curated VQA training data, yet it robustly generalizes the visual graph comprehension capabilities to diagrams in current VQA benchmarks. We can observe a remarkable improvement of 4.32% on ScienceQA and 2.66% on MMBench. Furthermore, by leveraging the node description and number detection tasks in our graph comprehension fine-tuning, we explore the impact of `GraphVis` on object hallucinations. Through evaluations using POPE across its three scenarios (random, popular, and adversarial) we find that `GraphVis` effectively reduces object hallucinations in the LVLM, enhancing both the accuracy and the F1 score in determining whether an object is present in an image. This results in an average improvement of 1.09%. Additionally, we note that differences exist between our visualized knowledge graphs and the graph images in these VQA benchmarks in terms of visual clarity, graph layout, information density, and domain knowledge. Despite these disparities, the model consistently shows improvements across various distinct benchmarks and demonstrates robust generalization capabilities, transitioning effectively from abstract graph structures to real-world images.

Table 2: VQA performance of `GraphVis` based on `llava-v1.6-mistral-7b`.

| Model | ScienceQA Img-Acc | MMBench Overall | POPE-ran | | POPE-pop | | POPE-adv | |
|---|---|---|---|---|---|---|---|---|
| | | | Acc | F1 | Acc | F1 | Acc | F1 |
| Base LVLM | 68.86 | 63.75 | 88.56 | 87.65 | 87.73 | 86.53 | 86.47 | 85.37 |
| GraphVis | **73.18**(+4.32) | **66.41**(+2.66) | **89.73** | **89.12** | **88.73** | **87.89** | **87.07** | **86.32** |

## 6 Ablation Study

In this section, we conduct further ablation studies to explore the different variants of `GraphVis` to illustrate the significance of the components within our method design.

**Curriculum fine-tuning.** `GraphVis` emphasizes a curriculum fine-tuning scheme, initially training the model on fundamental visual graph concepts, such as node number and node degrees. Only after mastering these basic comprehension tasks does the model advance to train on the more complex

reasoning tasks which require leveraging visual graphs to answer relevant questions. Here, we evaluate the impact of task sequencing in fine-tuning by comparing the standard `GraphVis` with a variant that jointly fine-tunes across the mixed data, encompassing both image comprehension and reasoning tasks. Additionally, we categorize image comprehension tasks into two distinct groups for more fine-grained curriculum fine-tuning:

- **OCR** tasks, including node description and triple listing.
- **Graph** tasks, including node degree detection, highest node degree detection, and node/edge number detection.

Figure 7 presents the CSQA performance results for each fine-tuning strategy of `GraphVis`. Although `GraphVis` generally enhances performance across the different schemes, improvements are notably less significant when tasks are jointly trained. The curriculum-based approach yields an additional gain of $4.51\%$ over joint fine-tuning. However, the benefits of more detailed fine-tuning appear minimal. Initiating fine-tuning with OCR tasks, followed by graph tasks and subsequent QA reasoning, leads to a marginal increase of $0.24\%$. Conversely, reversing the order of these detailed tasks results in a performance decline of $1.56\%$. These findings indicate that optimal fine-tuning involves separating initial image comprehension stages from subsequent reasoning tasks.

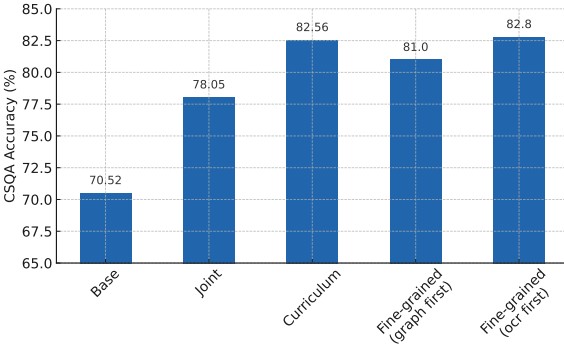

Figure 7: Comparison between different fine-tuning schemes with `GraphVis` on CSQA.

**Combination with prompting.** As `GraphVis` is intrinsically compatible with prompting methods like KAPING, we explore the integration of the two methods, as demonstrated in Table 3. Firstly, due to the strong performance of current LLMs, incorporating KAPING into the base model results in a performance decline of $2.87\%$, a trend consistent with results reported in the previous study (Tian et al., 2024). Such degradation can be attributed to both the prolonged context and the fact that the original model was not adept at understanding graph structures. Meanwhile, for `GraphVis` with joint fine-tuning that understands the graph structure as well as the triplet format through the task of triple listing, an improvement of $0.66\%$ is observed. However, for `GraphVis` with curriculum fine-tuning that has more effectively learned the visual graph, the addition of KAPING prompts appears to be redundant and causes a minor degradation of $0.82\%$.

Table 3: Performance of `GraphVis` based on `llava-v1.6-mistral-7b` with or without the prompting from KAPING.

|  | Original | w/ KAPING |
|---|---|---|
| Base LVLM | 70.52 | $67.65_{(-2.87)}$ |
| `GraphVis` (Joint) | 78.05 | $78.71_{(+0.66)}$ |
| `GraphVis` | 82.56 | $81.74_{(-0.82)}$ |

**Performance on graph comprehension task.** In Table 4, we further evaluate the LVLM on the graph comprehension tasks we defined, both before and after training on the synthetic tasks. To ensure a fair comparison, we utilized synthetic images from the test data of CSQA to construct a test set. The accuracy for each individual task is reported. We implement exact matching in determining answer accuracy, which, while strict, provides insight into performance gains and error sources. We observed that graph comprehension tasks are essentially difficult for the LVLM, as such images and tasks are scarce in its pre-training and fine-tuning data. On tasks such as triple listing, it almost cannot fulfill the task. For an output example: "*Based on the image provided, the graph appears to represent a network or a system with nodes (blue circles) and edges (black lines) connecting them. To list all the triples in the graph, I'll describe each triple as a sequence of three nodes in the*

*graph, which are connected by edges. Here are the triples in the graph: 1. (node1, node2, node3) 2. (node2, node3, node4)...*" Since these preliminary tasks were considered a warm start for the model to learn grounding its reasoning on graph images, we only fine-tuned on these tasks for one epoch. Nevertheless, we observed a notable gain across all tasks after just one epoch of fine-tuning.

Table 4: Performance of LLaVA-v1.6 before and after fine-tuning on each graph comprehension task. N. denotes node and E. denotes Edge. For node description and triple listing, we consider the average accuracy of each test example. We use exact matching to determine the accuracy, which may be a stricter evaluation.

| Model | N. description | N. degree | Highest N. degree | N. number | E. number | Triple listing |
|---|---|---|---|---|---|---|
| Original | 1.4 | 15.3 | 3.3 | 16.7 | 9.7 | 0.6 |
| After fine-tuning | $12.8_{(+11.4)}$ | $27.0_{(+11.7)}$ | $11.6_{(+8.3)}$ | $27.5_{(+10.8)}$ | $16.2_{(+9.7)}$ | $8.2_{(+7.6)}$ |

**Qualitative example.**   In Figure 8, we provide a specific example of the model generations for the VQA task ScienceQA. The displayed question fundamentally requires the model to traverse through the food web from a starting point, following the directed arrows, and match the target node names with the provided options. The original model failed to complete this task successfully. However, with `GraphVis`, the model's ability to handle such image data improved significantly, resulting in a correct answer.

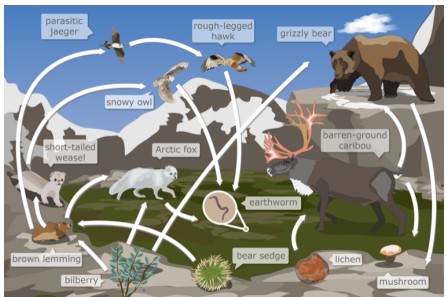

**Query**: Context: Below is a food web from a tundra ecosystem in Nunavut, a territory in Northern Canada. A food web models how the matter eaten by organisms moves through an ecosystem. The arrows in a food web represent how matter moves between organisms in an ecosystem.
Which of these organisms contains matter that was once part of the lichen?
A. mushroom    B. short-tailed weasel    C. brown lemming
D. rough-legged hawk    E. bilberry
Answer with the option's letter from the given choices directly.

**Base (LLaVA-v1.6 7B)**: C        **GraphVis (LLaVA-v1.6 7B)**: A

Figure 8: Example of model output on ScienceQA (VQA task). Note that after fine-tuning the base LVLM with GraphVis on CSQA with synthetic KG images, the model can successfully traverse the graph to locate the correct answer.

# 7   Conclusion

In conclusion, we proposed `GraphVis`, a new approach to integrate structured knowledge from KGs with LLMs through the visual modality. By preserving the intricate graph structure and employing a curriculum fine-tuning scheme, our method not only enhanced LLMs' ability to comprehend and reason over KG data to enhance its response to textual QAs but also significantly improves performance across several VQA benchmarks. `GraphVis` leveraged the strengths of both textual, visual and KG data, reducing factual inaccuracies and hallucinations typical in LLM outputs. The promising results achieved on multiple benchmarks underscore the potential of `GraphVis` to set a new approach of utilizing data from the KG modality and enhancing a model's performance in the cross-modal fashion.

**Limitations and future work.** Firstly, we acknowledge the limitation induced by compute resources that our experiments are done on 7B models with LoRA fine-tuning. If compute resource permits, it is interesting to scale up the experiments with larger models and full fine-tuning. Another limitation is the size of the retrieved subgraph, for which we considerd a 2-hop subgraph to ensure that the visualization is not too complicated for the vision model to recognize. Extending from our current method, interesting future work includes exploring how different visualizations may influence the effectiveness of `GraphVis`. Additionally, instead of following the previous retrieval methods, it would be valuable to investigate better subgraph retrieval techniques and integrate them into the learning process. Lastly, while we used ConceptNet as an example KG to enhance commonsense reasoning, there are numerous other KGs available. It is crucial to explore the generalizability of `GraphVis` to adapt to new KGs. Furthermore, it is possible to obtain multiple relevant subgraphs for a given question from different KG sources. An open problem remains on how to leverage multiple KG subgraphs for enhanced reasoning in LLMs.

## Acknowledgments

We sincerely thank the anonymous reviewers for their helpful comments. The work is partially supported by DARPA HR0011-24-9-0370, NSF 2200274, 2106859, 2312501, NIH U54HG012517, U24DK097771, and Optum AI. The views and conclusions contained in this paper are those of the authors and should not be interpreted as representing any funding agencies.

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

# A  Experiment Details

**Visual graph comprehension data.**    We created a pool of five prompts for each of the task in visual graph comprehension, where the answers can be automatically extracted. For node description, we have

- "List all nodes of the graph shown in the image."
- "Provide the names of all nodes displayed in the graph image."
- "Can you name all the nodes shown in the graph image?"
- "Identify all the vertices in the diagram of the graph provided."
- "Detail all the vertices from the graph depicted in the image."

For highest node degree detection, we have

- "Name one of the node with the highest degree in the graph. And what is its degree?"
- "Identify one of the node that has the most connections in the graph and specify its degree."
- "Can you tell me which node (name one) has the highest degree in this graph and what that degree is?"
- "Provide the name and degree of the node with the most connections in the graph."
- "Which node in the graph has the greatest number of connections, and what is that total?"

For node degree detection, we have

- "What is the degree of the node with the name "node"?",
- "What is the degree of the node labeled "node"?",
- "Can you tell me the degree of the node named "node"?"
- "What is the total number of connections that the node "node" has?"
- "How many connections does the node "node" have?"

For node number detection, we have

- "How many nodes are there in the graph?"
- "What is the total number of nodes in the graph?"
- "Can you tell me how many nodes are in the graph?"
- "What is the total number of vertices in the graph?"
- "How many vertices are there in the graph?"

For edge number detection, we have

- "How many edges are there in the graph?"
- "What is the total number of edges in the graph?"
- "Can you tell me how many edges are in the graph?"
- "What is the total number of connections in the graph?"
- "How many connections are there in the graph?"

For triple listing, we have

- "List all the triples in the graph."
- "Provide all the triples in the graph."
- "Can you list all the triples in the graph?"
- "Detail all the triples in the graph."
- "Enumerate all the triples in the graph."

**Fine-tuning.**    We train 1 epoch for both part of the fine-tuning process. We present the fine-tuning hyperparameters of GraphVis in Table 5.

Table 5: Fine-tuning hyperparameters.

| | |
|---|---|
| lora_r | 128 |
| lora_alpha | 256 |
| lora_target | all |
| Learning rate | 1e-7 |
| Optimizer | AdamW |
| Global batch size | 4 |
| gradient_accumulation_steps | 1 |
| weight_decay | 0 |
| warmup_ratio | 0.03 |
| lr_scheduler_type | cosine |
| image_aspect_ratio | pad |
| group_by_modality_length | True |
| model_max_length | 2048 |
| mm_projector_lr | 2e-5 |
| mm_projector_type | mlp2x_gelu |

**Evaluation.** We use the same evaluation scripts provided by LLaVA (Liu et al., 2023a) for all evaluations performed in this paper. We note that the new evaluation scripts (prompts) used to report the newest results of LLaVA-v1.6 are not released yet, which may cause minor differences in evaluation results of the original model compared to their reported values. Nevertheless, we use the same evaluation scripts throughout the paper to ensure fairness in comparison.

**Compute resources.** Experiments of this paper were all conducted on NVIDIA RTX A6000 GPU clusters. The fine-tuning of LLaVA v1.5 (7B) on the visualized subgraphs takes approximately 3 hours on 4 GPUs. The time span for evaluations on the different benchmarks range from 0.5 to 8 hours using 1 GPU, depending on the varying size of the dataset.

**Additional Experiment Results** In Table 6, we include the additional results on ScienceQA as one of the VQA tasks from either doing a curriculum fine-tuning or simply joint fine-tuning on the curated synthetic data. As indicated by the results, curriculum learning transfers to these VQA tasks as well. In Figure 9, we investigate the influence of image quality for the synthetic visual graphs used for

Table 6: Performance of LLaVA-v1.6 on ScienceQA compared with GraphVis and GraphVis (joint fine-tuning).

| | ScienceQA (%) |
|---|---|
| Base LVLM | 68.86 |
| GraphVis (Joint) | 71.94 |
| GraphVis | **73.18** |

training. It is generally observed in VQA tasks that images with lower resolution can lead to degraded performance, as these images are considered as "corrupted" and often leads to object hallucinations. For the QA tasks that we considered in our evaluation, we conducted additional experiments using graph images with smaller sizes and consequently lower resolutions (50x50).

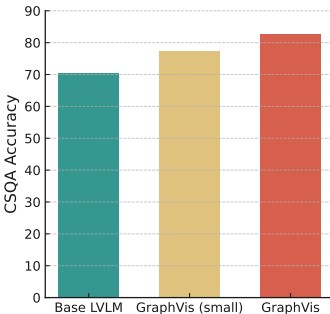

Figure 9: Effect of lower resolution graph images.

# B Broader Impact

By leveraging factual information from KGs, `GraphVis` aims to mitigate inaccuracies in the reasoning process and effectively reduced hallucinations in model outputs. This approach aims for a more accurate and reliable model, contributing positively in social impact by providing a more trustworthy and accountable AI model. The improved accuracy and reliability of `GraphVis` can enhance user trust in AI applications, especially in critical areas such as healthcare, education, and legal advice.

Meanwhile, there are potential negative societal impacts of enhanced LVLMs capabilities. As `GraphVis` increases the effectiveness of these models, there is a risk of misuse in ways that could harm privacy and fairness. For instance, more advanced LVLMs could be exploited to generate misleading or deceptive content or amplify biases present in the underlying data, leading to unfair outcomes. To address these concerns, it is crucial to ensure transparency in how the models are trained and used, incorporating bias detection and mitigation strategies.

