# OpenReview forum: "GraphVis: Boosting LLMs with Visual Knowledge Graph Integration"
_NeurIPS.cc/2024/Conference — NeurIPS 2024 poster_

### Official Review · Reviewer_Tf8h · 2024-07-04

**Soundness:** 3
**Presentation:** 3
**Contribution:** 3
**Rating:** 6
**Confidence:** 3

**Summary:**

- This paper proposes an instruction tuning method with visual knowledge graph to enhance the large vision language models with external knowledge and perform better on QA tasks.

**Strengths:**

- It is a new idea that organize the external knowledge as an image to enhance the LVLMs.

**Weaknesses:**

- Only LLaVA-v1.6-Mistral is employed as a LVLM backbone in the experiments. As there are also many other LVLMs, I think more experiments should be conducted on diverse LVLMs to demonstrate the effectiveness of GraphVis.
- The visual graph understanding ability is not evaluated seperately. Only the final performance of VQA tasks are reported in the experiments. As a two-stage training framework, I think the Visual Graph Comprehension Fine-tuning should be also evaluated with suitable metrics.
- How can it be proved with some examples that the Visual Graph Comprehension Fine-tuning really works?

**Questions:**

I hope authors could consider my comments and give me a response.

**Limitations:**

There is a limitation section in the paper.

---

> ### Author Rebuttal · Authors · 2024-08-07
>
> We're grateful for your support and helpful feedback. Please find our response below for the questions raised in the review and additional experiments.
>
> ---
>
> **Q1**. More experiments should be conducted on diverse LVLMs to demonstrate the effectiveness of GraphVis.
>
> **A1**. Thank you for your suggestion. In our limitations discussion, we highlighted that scaling up experiments with larger models would be an interesting direction if compute resources permit. In response to the raised point, we have added experiments on the CSQA task using LLaVA-v1.5 (Vicuna-7B). This model has a different LLM backbone and pre-training process compared to the one initially used in our study.
>
> | Method | CSQA |
> | :--------: | :--------: |
> | Base LVLM | 68.1  |
> | GraphVis |79.9  |
>
> ---
>
> **Q2**. Evaluate the model’s visual graph understanding ability separately.
>
> **A2**. Thank you for your suggestion. In Table 1 of the attached PDF, we further evaluated LVLM on these graph comprehension tasks, both before and after training on the synthetic tasks. To ensure a fair comparison, we utilized synthetic images from the test data of CSQA to construct a test set. The accuracy for each individual task is reported. Due to time constraints, we implemented exact matching in determining answer accuracy, which, while strict, provides insight into performance gains and error sources.
>
> We observed that graph comprehension tasks are essentially difficult for the LVLM, as such images and tasks are scarce in its pre-training and fine-tuning data. On tasks such as triple listing, it almost cannot fulfill the task. For an output example:
>
> "Based on the image provided, the graph appears to represent a network or a system with nodes (blue circles) and edges (black lines) connecting them. To list all the triples in the graph, I'll describe each triple as a sequence of three nodes in the graph, which are connected by edges. Here are the triples in the graph: 1. (node1, node2, node3) 2. (node2, node3, node4)..."
>
> Since these preliminary tasks were considered a warm start for the model to learn grounding its reasoning on graph images, we only fine-tuned on these tasks for one epoch. Nevertheless, we observed a notable gain across all tasks after just one epoch of fine-tuning.
>
> ---
>
> **Q3**. Examples to show the effectiveness of GraphVis.
>
> **A3**. Thank you for your suggestion. In Figure 4 of our [attached one-page pdf](https://openreview.net/attachment?id=qU5a2KzdFg&name=pdf), we provided a specific example in the mode generations for the VQA task ScienceQA. The question fundamentally requires the model to traverse through the food web from a starting point, following the directed arrows. It must identify the potential nodes connected to the starting point in a certain direction and match the node names with the provided options. The original model failed to complete this task successfully. However, with  GraphVis, the model's ability to handle such image data improved significantly, resulting in a correct answer.

---

> > ### Comment · Reviewer_Tf8h · 2024-08-08
> > **Rating Change**
> >
> > Thank you for your response. I have change my rating from 5 to 6.

---

> > > ### Author Response · Authors · 2024-08-08
> > > **Thank you**
> > >
> > > Thank you for your prompt response and positive feedback on our rebuttal!

---

### Official Review · Reviewer_omgG · 2024-07-07

**Soundness:** 3
**Presentation:** 3
**Contribution:** 3
**Rating:** 6
**Confidence:** 4

**Summary:**

The paper presents a method to improve large vision language models (LVLMs) by integrating knowledge graphs (KGs) visually. The approach, GraphVis, uses LVLMs to understand KGs through image visualizations, enhancing comprehension and reasoning. It employs a curriculum fine-tuning strategy, starting with simple graph features and moving to complex QA tasks. Experiments show significant performance gains in both textual QA and zero-shot VQA, outperforming existing methods with efficient parameter training. The paper also addresses potential societal impacts and future work.

**Strengths:**

1. The paper introduces a novel approach, GraphVis, which effectively integrates structured knowledge from knowledge graphs into large visual language models using a visual modality.
2. GraphVis employs a unique, sequential curriculum fine-tuning scheme that progressively trains the model on basic graph features before moving to more complex reasoning tasks.
3. The paper demonstrates that GraphVis not only improves textual question-answering performance but also enhances zero-shot visual question-answering capabilities.

**Weaknesses:**

1. While the paper demonstrates the effectiveness of GraphVis using ConceptNet, it may not be clear how well the approach generalizes to other knowledge graphs with different domains. Further research would be needed to confirm its effectiveness across various KGs.
2. What is the diversity of question-answer pairs used for visual understanding finetuning? Could training on overly monotonous datasets lead to overfitting, and would this affect downstream tasks?
3. Even if two graphs have identical structures, they can present different visualization results through changes in the positions of nodes, edges, and other elements. Would this affect the training of the model?
4. In Table 1, the comparison seems unfair for baseline methods. It could be better if authors provide the zero-shot performance on CSQA and OBQA by training on other available datasets.

**Questions:**

Please refer to weaknesses.

---

> ### Author Rebuttal · Authors · 2024-08-07
>
> Thank you for your support and constructive feedback. Please find our detailed response below for the questions raised in the review.
>
> ---
>
> **Q1**. Further research would be needed to confirm its effectiveness across various KGs.
>
> **A1**. Thanks for raising this important aspect. As we have also mentioned in our conclusion paragraph on limitations and future work, it is indeed crucial to explore the generalizability of GraphVis across different KGs. While these investigations are beyond the scope of our current study, we will further emphasize this future research direction and outline our plans for future investigations in our revised manuscript.
>
> ---
>
> **Q2**. What is the diversity of question-answer pairs used for visual understanding finetuning? Could training on overly monotonous datasets lead to overfitting, and would this affect downstream tasks?
>
> **A2**. Thank you for pointing out this. The synthetic QA data may indeed result in an overly monotonous dataset, and therefore we employed different questions for each synthetic task to increase the diversity. Moreover, as we discussed in line 197-201, we incorporated real QA data for finetuning, which further mitigates this potential issue.
>
> To evaluate the impact of question format diversity on model performance, we conducted experiments using only one of the question formats for each synthetic task, as compared to the 5 formats used in our paper. We note that the 1 format is contained in the 5 formats for a fair comparison. The results are summarized in Table 4 of [our attached pdf](https://openreview.net/attachment?id=qU5a2KzdFg&name=pdf).
>
> ---
>
> **Q3**. Visualization may change even when the structure of the graph is identical. Would this affect the training of the model?
>
> **A3**. Thank you for raising this question. We acknowledge that different visualizations can exist for the same graph structure. In our study, we generated random visualizations of the graphs to capture an average performance across these variations.
>
> To address the potential effects of different visualizations, we conducted an additional experiment using another set of randomly generated images with different visualization colors and shapes for the CSQA dataset (as shown in Table 5 of our attached pdf). The results of this experiment will be included in our revised manuscript to show the robustness of our result.
>
> We further recognize that leveraging the diversity in graph images for the same graph structure is an interesting future direction. Utilizing this diversity could further enhance the model's ability to generalize and improve its robustness to different visual representations. We will add this aspect in the discussion on future research.
>
> ---
>
> **Q4**. It could be better if authors provide the zero-shot performance on CSQA and OBQA by training on other available datasets.
>
> **A4**. We first clarify that the fine-tuning baselines that we compare with (e.g. GNP) similarly use the training data from CSQA and OBQA. To provide stronger and more comprehensive baselines, we conduct experiments with fine-tuning on the same training data without any additional KG information, and fine-tuning with KAPING prompting. Results are shown in Table 2 of our attached pdf, which continue to demonstrate the effectiveness of our method. We will include these two additional baselines in our Table 1 in our revision.
>
> Regarding the generalizability of our results, we would like to point out that our setting for VQA is indeed zero-shot. With fine-tuning on these synthetic graph images and textual QAs, the LVLM exhibited notable improvement on zero-shot tasks such as ScienceQA and MMBench, which also contains data of graph structures. These results highlight the successful transfer and generalization capabilities of our approach across different datasets and tasks.
>
> ---
>
> Thank you again for your helpful comments. We hope that our clarifications and additional experiments address the raised concerns.

---

> > ### Comment · Reviewer_omgG · 2024-08-12
> > **Thank you for the response.**
> >
> > The responses addressed my concerns. I have changed my rating.

---

> > > ### Author Response · Authors · 2024-08-12
> > > **Thank you**
> > >
> > > Thank you for getting back to us and for the positive feedback on our rebuttal!

---

> ### Author Response · Authors · 2024-08-12
>
> Dear reviewer omgG,
>
> Thank you again for your support and valuable feedback. We appreciate your insights and hope that we have adequately addressed your questions. Specifically,
>
> 1. Exploring various KGs: We agree this is an important area for future work and will include a discussion in our revision.
> 2. Additional experiments: We've provided new results on:
>     - The effect of diverse question formats (Table 4 in the [attached PDF](https://openreview.net/attachment?id=qU5a2KzdFg&name=pdf))
>     - Different visualizations for synthetic graph images (Table 5 in the [attached PDF](https://openreview.net/attachment?id=qU5a2KzdFg&name=pdf))
> 3. Generalizability: We included additional baselines that were similarly fine-tuned and tested on the same data (Table 2 in the [attached PDF](https://openreview.net/attachment?id=qU5a2KzdFg&name=pdf)). Specifically regarding generalization, we clarified that our VQA tasks are performed in a zero-shot setting. Further explanation of our contribution regarding VQA is presented in [Global A3](https://openreview.net/forum?id=haVPmN8UGi&noteId=qU5a2KzdFg).
>
> We hope these responses and clarifications have been helpful. If you have any further questions about our rebuttal, we're happy to provide additional information or clarification. We sincerely appreciate the time and effort you've invested in reviewing our work!

---

### Official Review · Reviewer_2Am7 · 2024-07-12

**Soundness:** 3
**Presentation:** 3
**Contribution:** 2
**Rating:** 6
**Confidence:** 4

**Summary:**

GraphVis introduces a novel method for integrating knowledge graphs (KGs) into large language models (LLMs) by preserving the graph structure through the visual modality. Utilizing Large Vision Language Models (LVLMs) and a curriculum fine-tuning scheme, GraphVis enhances both textual QA and VQA performance, demonstrating significant improvements over existing KG-enhanced LLM methods.

**Strengths:**

1. The use of visual representations to preserve the intricate structure of KGs is a novel approach, addressing limitations of linearized text triples and improving the expressiveness of structured data integration.
2. The two-phase curriculum fine-tuning, starting with graphical feature recognition and progressing to reasoning tasks, is a technically-sound strategy.
3. The paper provides extensive evaluations across commonsense reasoning QA and VQA benchmarks, showcasing substantial performance gains.

**Weaknesses:**

1. The paper does not discuss how the properties of the generated graph images, such as size and resolution, affect the model's performance. Understanding these factors is crucial for replicating and optimizing the method.
2. While the paper claims performance improvements, it is important to confirm whether this cross-modal methodology is a novel approach. A comparison with existing methods and a discussion on how GraphVis advances the current state-of-the-art would be beneficial.
3. The proposed method seems to lack significant technical contributions. The approach primarily leverages existing techniques (e.g., curriculum fine-tuning and visual graph generation) without introducing substantial innovations.

**Questions:**

As mentioned in Weaknesses.

**Limitations:**

Yes

---

> ### Author Rebuttal · Authors · 2024-08-07
>
> We appreciate your support and suggestions, for which we have included additional experiments accordingly. We hope our explanations below answer your questions and provide more clarity.
>
> ---
>
> **Q1**. How the properties of the generated graph images, such as size and resolution, affect the model’s performance?
>
> **A1**. Thank you for your suggestion. It is generally observed in VQA tasks that images with lower resolution can lead to degraded performance, as these images are considered as "corrupted" and often leads to object hallucinations. For the QA tasks that we considered in our evaluation, we conducted additional experiments using graph images with smaller sizes and consequently lower resolutions (50x50). The results are summarized in Figure 3 of our [attached one-page pdf](https://openreview.net/attachment?id=qU5a2KzdFg&name=pdf).
>
> From the figure, we can observe that reducing the size and resolution of the graph images (GraphVis (small)) leads to a decrease in performance compared to the standard GraphVis setup. This indicates that higher resolution graph images are crucial for the model to accurately comprehend and utilize the visual information encoded in the graph images as well.
>
> Lastly, we emphasize that image size and resolution can be considered as hyperparameter choices. This does not affect our main contribution, which is demonstrating that graph images are a more effective means of conveying useful graph information compared to verbalization.
>
> ---
>
> **Q2**. While the paper claims performance improvements, it is important to confirm whether this cross-modal methodology is a novel approach.
>
> **A2**. Thank you for pointing out the importance of highlighting the novelty of GraphVis. The contribution of GraphVis is two-fold.
> 1. **Novel use of visual modality for KG-Enhanced LLMs**: GraphVis is the first to employ visual modality for KG-enhanced LLMs, leveraging graph visualization to bridge the gap between structured KG data and multimodal LLM processing capabilities.
> 2. **Utilization of KG and textual data in fine-tuning LVLMs**: GraphVis uniquely proposes the use of KG and textual data to fine-tune LVLMs by leveraging synthetic graph images and textual QA datasets.
>
> We recognize there may be some misunderstanding regarding our second contribution. To clarify, the VQA tasks considered in our paper were performed in a zero-shot setting. Our primary contribution lies in demonstrating the potential of utilizing vast training data from the text-only domain and generating synthetic images with graph structures to enhance the LVLM’s understanding of images that have underlying graph structures.
>
> To provide further clarity, the current LVLMs (e.g. LLaVA-v1.6) are pre-trained and fine-tuned on large corpus of vision-language instruction-following data. This corpus includes data curated from various VQA training datasets, human annotations, and GPT-4V generations. Obtaining such training data for LVLMs, however, is considerably expensive as it involves data from different modalities. For instance, generating 6k image descriptions with 1k tokens per output using GPT-4V would cost approximately $200. Therefore, researchers have been exploring ways of generating synthetic data to further improve these LVLMs [1-3].
>
> Our major contribution here is to offer a new perspective on gathering fine-tuning data to enhance LVLMs. Specifically, we propose that pure textual data can be combined with relevant synthetic graph images derived from KGs to improve the LVLM’s capability in image comprehension and reasoning. This approach is particularly beneficial for images with graph structures, which are relatively scarce.
>
> We will include and highlight the above discussion in our revised manuscript to provide more clarity on our contributions.
>
> ---
>
> **Q3**. Further discussion on technical contribution.
>
> **A3**. Thank you for your interest in the technical contributions of our work. As we addressed in our response to Q2, there may have been some misunderstanding regarding our contributions. Here, we clarify and elaborate on the technical advancements introduced by GraphVis.
>
> Our method is novel in advancing cross-modal improvements, which has not been explored for the three modalities that we consider, through two key mechanisms:
>
> Visual modality for KG-Enhanced LLMs: GraphVis is the first approach to integrate visual representations of KGs within the processing framework of KG-enhanced LLMs.
> Synthetic graph images for fine-tuning LVLMs: By combining textual data with synthetic visual graphs, we enhance the LVLM's ability to process and reason about graph-structured information, leading to improved performance in tasks that involve images with underlying graph structures. This approach is particularly valuable given the scarcity of real-world images with graph structures.
>
> We emphasize that efficient and affordable data curation for the fine-tuning of LLMs and LVLMs is one of the most important technical directions for advancing their performance. A significant body of work focuses on synthetic data generation (unimodal [4-6] or multimodal [1-3]) rather than redesigning or modifying the architecture of the large models. Our work contributes to this direction by providing a new method for leveraging data from an unused domain and generating synthetic multimodal data that enhances LVLM capabilities.
>
> ---
>
> [1] Aligning modalities in vision large language models via preference fine-tuning.
>
> [2] Enhancing large vision language models with self-training on image comprehension.
>
> [3] Understanding alignment in multimodal LLMs: a comprehensive study.
>
> [4] Self-rewarding language models.
>
> [5] Beyond Human Data: Scaling Self-Training for Problem-Solving with Language Models.
>
> [6] Scaling relationship on learning mathematical reasoning with large language models.

---

> > ### Comment · Reviewer_2Am7 · 2024-08-09
> > **Reply to authors**
> >
> > Thanks for your response. I have updated my score.

---

> > > ### Author Response · Authors · 2024-08-09
> > > **Thank you**
> > >
> > > Thank you for your timely and encouraging feedback on our rebuttal!

---

### Official Review · Reviewer_GUVR · 2024-07-13

**Soundness:** 2
**Presentation:** 3
**Contribution:** 3
**Rating:** 5
**Confidence:** 3

**Summary:**

The paper introduces GraphVis, a novel approach that enables Large Vision Language Models (LVLMs) to reason about visual knowledge graphs for QA tasks.  Unlike previous methods that either input knowledge graph (KG) triplets directly to LLMs or use graphical neural networks to capture structured representations, GraphVis instead represents the graph visually as images of nodes and edges. This inherently introduces an auxiliary OCR task to parse the image of visual knowledge graph, and then the model needs to take the graphical image as input to answer commonsense questions. The knowledge graphs are derived from ConcetNet and relevant nodes and edges are extracted based on the question and answer choices by using off the shelf parser. The authors demonstrate that finetuning LLaVA 1.6 on textual commonsense questions without paired image but instead with the retrieved visual knowledge graphs, leads to improvements in CSQA, as well as zero-shot performance in VQA tasks such as MMBench and ScienceQA.

**Strengths:**

**Originality and Significance:**

While knowledge augmented LLMs have been explored in prior work, GraphVis is the first to leverage the multimodal capabilities of LVLMs to explicitly understand and interpret the structured relationships visually represented in knowledge graphs (KGs). Incorporating tables, graph figures, and other structured representation as visual context has been investigated, but not specifically for knowledge graphs.

The paper demonstrates that training on text only QAs with synthetically generated knowledge graphs improves the zero-shot performance of visual QAs over the base model. This is quite a significant finding, and opens up more interesting applications of aligning text data with synthetic multimodal context for improving multimodal tasks.

Ablation studies thoroughly examine different strategies of training stages and the order of prompts for graph reasoning questions, and show that separating initial image comprehension stages from subsequent reasoning tasks leads to the best result.

**Clarity and Quality:**

The paper clearly shows how a visual graph is constructed and interpreted by LVLMs. It is easy for readers to follow their training stages. Example images of VQA tasks involving graphs help the readers to understand the transferability of comprehending synthetic KGs in a multimodal setup.

**Weaknesses:**

- Missing details and analysis of the synthetic visual graphs. Authors should show the statistics of retrieved subgraphs per question, including average number of nodes, degrees of freedom, and etc. Evaluation of the visual graph comprehension tasks should be included to measure how effectively LVLMs understand the graph structure accurately, and their main sources of error in graph comprehension.
- Unfair Comparison of GraphVis to zero shot approaches. KAPING is a zero-shot approach that involves no model training and only augments the knowledge directly in the input of LLM. GraphVis instead finetunes the model to understand the visual graph as context. A more fair comparison would be to either follow KAPING with finetuning by training **LVLMs with knowledge graph triplets**, or evaluate zero-shot performance of LVLMs with visual graph as input but no finetuning.
- Not enough evidence if the improvement comes from integrating the visual knowledge graph, or finetuning on the QA data.  The authors mostly compare GraphVis to the baseline LLaVA model trained on visual instruction tuning data. Since GraphVis additionally finetunes the base model with QA data, it is no surprise that the model outperforms the base model across the QA tasks. More appropriate baseline candidates are finetuning Mistral-7B LLM or LLaVA LVLMs with the QA data.
  - The authors should also include all ablation studies for VQA tasks, and not only for CSQA.
- Missing qualitative results of GraphVis vs baseline to show the benefits of visual graphs for VQA tasks.

**Questions:**

Questions and suggestions are derived from the weaknesses section.
1. Disentangle the benefits of finetuning with visual graphs vs on QA data. From the ablation studies, it is not clear if the model improves by comprehending and integrating the visual graphs, or by simply finetuning more on QA data.
2. Introduce more fair comparison of prior work. Authors should follow the finetuning adaptation of KAPING by train models with KG triplets, instead of visual graphs.
3. What are the main sources of error for graph comprehension tasks? It would be helpful  to identify what might be the bottleneck for understanding the graphical structure when KGs are presented as image.
4. Authors should present convincing qualitative results of their GraphVis model, not only on how the dataset is constructed.
5. What are the results if visual graph comprehension stage is omitted, and directly proceeds to KG-Enhanced QA Fine-tuning?

**Limitations:**

The authors adequately addressed the limitations.

---

> ### Author Rebuttal · Authors · 2024-08-07
>
> Thank you for your constructive feedback. We appreciate your recognition of the originality and significance of our work, and grateful for the positive feedback on the clarity and quality of our paper. Regarding the raised questions, please find our detailed response below with additional experiments and clarifications to potential misunderstanding. We organized the questions according to the order of the weaknesses section.
>
> ---
>
> **Q1**. Could the authors provide more details and analysis of the synthetic visual graphs? What are the performance and the main sources of error for graph comprehension tasks?
>
> **A1**. Thank you for raising these important questions. Please find our detailed response to each question in [global rebuttal](https://openreview.net/forum?id=haVPmN8UGi&noteId=qU5a2KzdFg) on Global A1 (with Figures 1 and 2 in [attached pdf](https://openreview.net/attachment?id=qU5a2KzdFg&name=pdf)) and Global A2 (with Table 1 in attached pdf).
>
> Concisely, the retrieved subgraphs (avg. 17 nodes, 25 edges) contain substantial information, but some can be overly complex. Graph comprehension tasks is challenging to LVLMs due to limited graph data in pre-training and fine-tuning. However, fine-tuning for just one epoch showed notable improvements.
>
> ---
>
> **Q2**. More discussions on the baselines. Adding KAPING with finetuning would be more comprehensive.
>
> **A2**. We aimed to provide a comprehensive overview and comparison by including the most recent baselines on KG-enhanced LLMs. While KAPING is a prompting method, we also included fine-tuning methods like KSL and GNP, as well as the larger models they selected (FLAN-T5 11B and GPT-3.5). In Table 1 of our paper, we also highlighted the differences between none fine-tuning and fine-tuning methods.
>
> In response to your suggestion, we included the additional baseline of fine-tuning LVLMs with KG triplets, to more comprehensively compare with KAPING. Following the original zero-shot setting, we maintained the top 10 retrieved triples and appended them to the question as the training data. The accuracy results of this comparison are presented in Table 2 of our attached pdf.
>
> As observed, KAPING with fine-tuning (KAPING w/ FT) shows an improvement over the base LVLM and zero-shot KAPING. Meanwhile, the vision-language model benefits more from a visual graph input than a linearized textual input. We also note that the computation of top-k embeddings among all retrieved triples was very time-consuming, while GraphVis does not require such computations.
>
> ---
>
> **Q3**. Disentangle the benefits of finetuning with visual graphs vs on QA data.
>
> **A3**. From our experiments in our previous response A2,fine-tuning with the additional visual information results in the best performances, while linearizing the retrieved knowledge subgraph into texts maintains helpful information but falls short in proving the useful structured information. In Table 1 of our attached pdf, we additionally add the baseline of fine-tuning on the QA training data without any additional KG information.
>
> We will include these two additional baselines in our revision.
>
> ---
>
> **Q4**. The authors should also include all ablation studies for VQA tasks.
>
> **A4**. As we do not retrieve KG subgraphs for the VQA tasks, it is not applicable to investigate our second ablation study “comparison with prompting”. However, in Table 3 of our attached pdf, we do include the additional results on ScienceQA as one of the VQA tasks from either doing a curriculum fine-tuning or simply joint fine-tuning on the curated synthetic data. As indicated by the results, curriculum learning transfers to these VQA tasks as well.
>
> Regarding the inapplicability of the second ablation study, we would like to further clarify our setting for VQA. The VQA tasks were done in the zero-shot setting and aimed to show the interesting benefits of fine-tuning LVLMs on the synthetic graph images and leveraging the existing textual QA data. As we highlighted in our paper, while current LVLMs are fine-tuned on human-labeled vision-language instruction data, images of complex graph structures are much more scarce compared to the many natural images, in addition to the scarcity of reasoning tasks designed for graph images. GraphVis highlights the potential to leverage textual data and KG images to improve the LVLM’s capability in reasoning with graph images. Therefore, we do not retrieve KG subgraphs for VQA tasks, but we consider that GraphVis provides a new perspective of data source for LVLMs. Our [Global A3](https://openreview.net/forum?id=haVPmN8UGi&noteId=qU5a2KzdFg) provides a more detailed explanation.
>
> ---
>
> **Q5**. Missing qualitative results of GraphVis vs baseline to show the benefits of visual graphs for VQA tasks.
>
> **A5**. Thank you for your suggestion. In Figure 4 of our attached one-page pdf, we provided a specific example in the mode generations for the VQA task ScienceQA. The question fundamentally requires the model to traverse through the food web from a starting point, following the directed arrows. It must identify the potential nodes connected to the starting point in a certain direction and match the node names with the provided options. The original model failed to complete this task successfully. However, with  GraphVis, the model's ability to handle such image data improved significantly, resulting in a correct answer.
>
> ---
>
> **Q6**. What are the results if visual graph comprehension stage is omitted, and directly proceeds to KG-Enhanced QA Fine-tuning?
>
> **A6**. We appreciate the suggestion to evaluate the impact of omitting the visual graph comprehension stage. In response, we conducted additional ablation studies to investigate this scenario. For the CSQA dataset, omitting the visual graph comprehension stage resulted in an accuracy of 77.5%, which underperforms the curriculum training result of 82.8%. We will add the comprehensive ablation study in our revision.

---

> ### Author Response · Authors · 2024-08-12
> **Inquiry for discussion**
>
> Dear reviewer GUVR,
>
> Thank you again for your constructive feedback and questions. We sincerely hope that our responses and clarifications have been helpful in addressing your questions and concerns. Specifically,
>
> 1. We provided additional statistics of the synthetic visual graphs ([Global A1](https://openreview.net/forum?id=haVPmN8UGi&noteId=qU5a2KzdFg)) as well as the model’s performance on each task ([Global A2](https://openreview.net/forum?id=haVPmN8UGi&noteId=qU5a2KzdFg)).
> 2. As suggested, we added more fine-tuning baselines including fine-tuning on textual QA only and fine-tuning with KG triples (Table 2 of [our attached pdf](https://openreview.net/attachment?id=qU5a2KzdFg&name=pdf))
> 3. We extended our ablation study to VQA tasks (Table 3 of [our attached pdf](https://openreview.net/attachment?id=qU5a2KzdFg&name=pdf))
> 4. We provided further explanation on the zero-shot setting of our VQA tasks and our corresponding contribution ([Global A3](https://openreview.net/forum?id=haVPmN8UGi&noteId=qU5a2KzdFg)).
> 5. We provided a specific example to show how GraphVis improves the LVLM on reasoning with graph images (Figure 4 of [our attached pdf](https://openreview.net/attachment?id=qU5a2KzdFg&name=pdf)).
>
> We would like to inquire if there are any questions about our rebuttal, for which we're happy to provide additional information and further clarifications. We truly appreciate the time and effort you’ve invested into reviewing our work!

---

> > ### Author Response · Authors · 2024-08-13
> >
> > Dear reviewer GUVR,
> >
> > Thank you again for taking the time to review our paper. We appreciate your detailed feedback and acknowledgement of the originality and significance of our work.
> >
> > In response to the feedback received, we have conducted additional experiments and provided detailed clarifications in our rebuttal. While most reviewers have responded positively to our revisions, we hope our detailed responses adequately address your concerns as well. We would appreciate your attention to our rebuttal and any further feedback you may have, as it will give us the opportunity to provide more details before the author-reviewer discussion session ends and help us continue to improve our work. Thank you for your valuable insights!

---

> > > ### Comment · Reviewer_GUVR · 2024-08-14
> > >
> > > Thank you for the response. I have raised the score.
> > > Please make sure to incorporate the results in the table of the attached pdf.
> > >
> > > For Table 1 on synthetic graph comprehension task, it would be good to see the performance of proprietary models (GPT4-o, Claude) or other current open-sourced models (InternVL, LLaVA-next) to make the table more complete.

---

> > > > ### Author Response · Authors · 2024-08-14
> > > >
> > > > Thank you for your reply and positive feedback! We appreciate your suggestions and will incorporate the additional experiment results into our revision. We will expand the results on the synthetic tasks with different model performances for better understanding of the tasks.

---

### Author Rebuttal · Authors · 2024-08-07

We sincerely thank all the reviewers for their insightful and encouraging feedback. We are grateful for the recognition of the novelty and significance of our work (Reviewer GUVR, 2Am7,omgG,Tf8h), extensive experiments and superior performance (Reviewer GUVR, 2Am7,omgG), clear writing flow (Reviewer GUVR), etc.

In response to the comments, we have provided additional experiments in the [attached one-page pdf](https://openreview.net/attachment?id=qU5a2KzdFg&name=pdf), including
1. **Distribution of Retrieved Subgraphs (Figures 1 and 2)**: With an average of 17 nodes and 25 edges, the retrieved 2-hop subgraphs contain substantial information. However, some subgraphs are overly complex, indicating room for improvement in pruning methods.
2. **Performance of LVLM on Synthetic Graph Comprehension Tasks (Table 1)**: These tasks are inherently challenging for LVLMs due to the scarcity of similar data in pre-training and fine-tuning stages. With fine-tuning on only one epoch, we observed a notable improvement.
3. **Additional Baselines (Table 2)**: Adding fine-tuned LVLMs with textual QA (Base w/FT) and textual KG triplets (KAPING w/FT) respectively. The visual graphs provided additional information and performance gains for vision-language models.
4. **Extending Ablation Study to VQAs (Table 3)**: We observed a similar pattern as in our ablation studies on QAs, where curriculum learning outperformed joint fine-tuning.
5. **Lower Resolution Images (Figure 3)**: By corrupting visual graph inputs (resizing), we observed a slight performance decay, consistent with universal observations that corrupted images lead to worse responses and more hallucinations.
6. **Number of Synthetic Question Formats (Table 4)**: Reduced diversity in synthetic question formats resulted in slight performance decay for GraphVis.
7. **Robustness to Different Visualizations of the Same Graph Architecture (Table 5)**: GraphVis demonstrated robustness to different visualizations.

For the most raised questions, please find our detailed response below.

---

**Global A1: statistics of the retrieved subgraphs**

We appreciate reviewers’ suggestions to include more detailed statistics on the synthetic visual graphs. In Figures 1 and 2 of our attached pdf, we provide the distribution of the number of nodes and edges in the retrieved subgraphs in CSQA. Specifically, we have one retrieved subgraph for each question, and the statistics for the retrieved subgraphs are:
- Average node number: 17.36
- Average edge number: 25.48
- Average node max degree: 7.82

---

**Global A2: evaluation on the synthetic graph comprehension tasks**

In Table 1 of the attached PDF, we further evaluated LVLM on these graph comprehension tasks, both before and after finetuning on the synthetic graphs' tasks. To ensure a fair comparison, we utilized synthetic images from the test data of CSQA to construct a test set. The accuracy for each individual task is reported. Due to time constraints, we report answer accuracy using exact matching , which, while strict, provides insight into performance gains and error sources.

We observed that graph comprehension tasks are essentially difficult for the LVLM, as such graph images and tasks are scarce in its pre-training and fine-tuning data. On tasks such as triple listing, it can hardly answer correctly. For an output example:

"Based on the image provided, the graph appears to represent a network or a system with nodes (blue circles) and edges (black lines) connecting them. To list all the triples in the graph, I'll describe each triple as a sequence of three nodes in the graph, which are connected by edges. Here are the triples in the graph: 1. (node1, node2, node3) 2. (node2, node3, node4)..."

Since these preliminary tasks were considered a warm start for the model to learn grounding its reasoning on graph images, we only fine-tuned on these tasks for one epoch. Nevertheless, we observed a notable gain ranging from 7.6\% to 11.7\% across all tasks after just one epoch of fine-tuning.

---

**Global A3: clarification on our contribution in VQA tasks**

The VQA tasks were done in the zero-shot setting and aimed to show the interesting benefits of fine-tuning LVLMs on the synthetic graph images and leveraging the existing textual QA data. As we highlighted in our paper, while current LVLMs are fine-tuned on human-labeled vision-language instruction data, images of complex graph structures are much more scarce compared to the many natural images. Also the reasoning tasks designed for complex graph images are very limited. GraphVis highlights the potential to leverage textual data and KG images to improve the LVLM’s capability in reasoning with graph images.

To further clarify, existing pre-training and fine-tuning corpus includes data curated from various VQA training datasets, human annotations, and GPT-4V generations. Obtaining such training data for LVLMs, however, is considerably expensive as it involves data from different modalities. For instance, generating 6k image descriptions with 1k tokens per output using GPT-4V would cost approximately $200. Therefore, researchers have been exploring ways of generating synthetic data to further improve the LVLMs [1-3].

Our major contribution here is to offer a new perspective on gathering fine-tuning data to enhance LVLMs. Specifically, we propose that pure textual data can be combined with relevant synthetic graph images derived from KGs to improve the LVLM’s capability in image comprehension and reasoning. This approach is particularly beneficial for images with graph structures, which are relatively scarce.

[1] Aligning modalities in vision large language models via preference fine-tuning.

[2] Enhancing large vision language models with self-training on image comprehension

[3] Understanding Alignment in Multimodal LLMs: A Comprehensive Study

---

We have also addressed the comments in each individual rebuttal.

---

### Decision · Program_Chairs · 2024-09-25

**Decision:**

Accept (poster)

**Comment:**

This paper introduces GraphVis, a novel method that integrates visual knowledge graphs with large vision language models (LVLMs) to enhance their reasoning capabilities on both textual and visual QA tasks. The reviewers raised concerns about the fairness of comparisons, the generalizability to other knowledge graphs, and the impact of visual graph comprehension (since this paper focus on fine-tuning open-source LLMs, comparison with closed-source LLMs are not necessary).
However, the rebuttal and discussion addressed these concerns well, with additional experiments, ablations, and clarifications provided, demonstrating the robustness and effectiveness of GraphVis. The reviewers unanimously provided positive scores following the discussion. The AC also believes the paper makes a unique and solid contribution to integrating knowledge graph into LLMs, which may inspire further large-scale training of LLMs/VLMs. The authors are encouraged to incorporate the additional results and clarifications and further address the comments in the reviews in camera ready.